# Planning with Reasoning using Vision Language World Model

## Abstract

Effective planning in the physical world requires strong world models, but world models that can understand and reason about high-level actions with semantic and temporal abstraction remain largely underdeveloped. We introduce the **Vision Language World Model (VLWM)**, a foundation model trained for language-based world modeling on natural videos. Given visual observations, the VLWM first infers the overall goal to be achieved then predicts a trajectory composed of interleaved actions and world state changes. These targets are extracted by iterative LLM SELF-REFINE conditioned on compressed future observations represented by TREE OF CAPTIONS. The VLWM learns both an action policy and a dynamics model, which respectively facilitates reactive **system-1** plan decoding and reflective **system-2** planning via cost minimization. The cost evaluates the semantic distance between the hypothetical future states given by VLWM roll-outs and the expected goal state, and is measured by a critic model that we trained in a self-supervised manner. The VLWM achieves state-of-the-art Visual Planning for Assistance (VPA) performance on both benchmark evaluations and our proposed PLANNERARENA human evaluations, where system-2 improves the Elo score by +27% upon system-1. The VLWM models also outperforms strong VLM baselines on RoboVQA and WorldPrediction benchmark.

## 1 Introduction

World models enable AI agents to optimize action plans internally instead of relying on exhaustive trial-and-error in real physical environments (LeCun, 2022; Ha & Schmidhuber, 2018; Fung et al., 2025), showing strong performance in planning across low-level, continuous control tasks such as robotic control (Oquab et al., 2023; Yang et al., 2024; Assran et al., 2025; Pan et al., 2025) and autonomous driving (Hu et al., 2023; Wang et al., 2024b). However, learning world models for *high-level* task planning – where actions involve semantic and temporal abstraction (Sutton et al., 1999; Chen et al., 2025) – remains an open challenge. Bridging this gap could unlock a wide range of practical applications, such as AI agents in wearable devices assisting humans in complex tasks and embodied agents capable of autonomously pursuing long-horizon goals.

Existing approaches fall short on producing such a high-level world model. Prompting-based practices (Hao et al., 2023; Tang et al., 2024; Wang et al., 2024a; Gu et al., 2024) is straightforward but inadequate as LLMs are not directly grounded in sensory experience. VLMs are primarily trained for visual perception rather than planning. Meanwhile, learning from simulated environments (Hafner et al., 2024; Wang et al., 2025b) cannot scale to diverse human activities in the real world. Existing world models trained from natural videos rely on generative architectures (*e.g.,* diffusion models) to output future observations (Yang et al., 2023b; Brooks et al., 2024; Agarwal et al., 2025b). Generative formulation is inadequate due to partial observation and uncertainty, as well as inefficient, capturing task-irrelevant details and imposing high computational costs for long-horizon roll-outs. These limitations highlight the need for world models that predict in *abstract representation spaces*, rather than raw pixels.

In this work, we propose to learn a world model that predicts action-state sequences from input video and a stated or inferred goal, and leverages natural language as its abstract world state representation. We introduce the **Vision Language World Model (VLWM)**, which perceives the environment through visual observations and predicts world evolution *using language-based representation* (Fig-

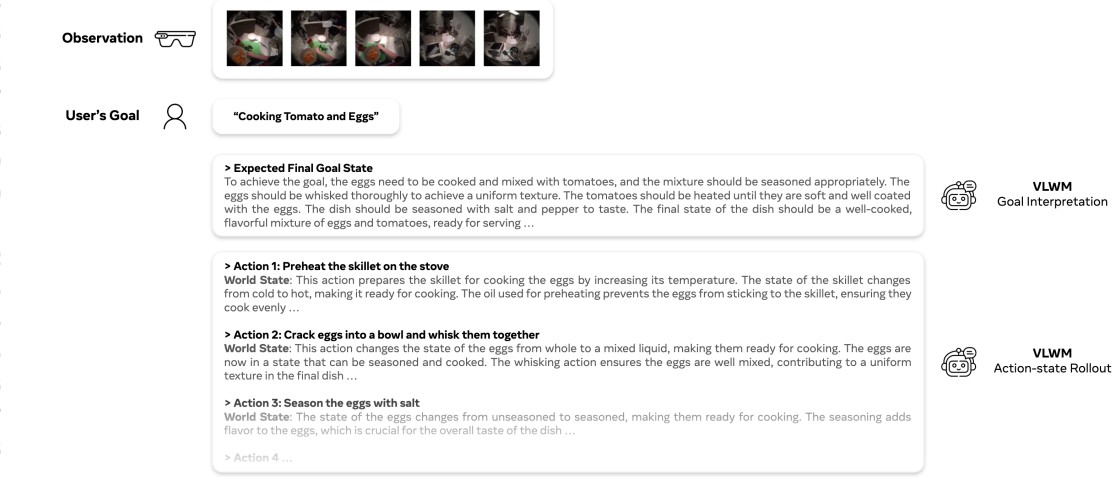

Figure 1: Example of a VLWM action-state trajectory given a video observation and a goal. VLWM can either generate a plan using one roll-out (system-1), or search over multiple actions by inferring the new world states and minimizing a given cost function (system-2).

ure 1). Language inherently provides semantic abstraction and is significantly more computationally efficient to generate compared to raw pixel observations. Language is also intuitive, interpretable, and enables seamless integration with prior knowledge and extensive engineering ecosystems developed for LLMs/VLMs. Compared to current LLMs/VLMs paradigms that primarily focus on perception (Cho et al., 2025), behavior cloning (SFT) (Zeng et al., 2023), or reinforcement learning with verifiable rewards (Shao et al., 2024), we propose to perform direct world modeling based on massive, uncurated videos, *i.e.,* reward-free offline data (Sobal et al., 2025). This approach has the advantage of being more scalable than the others.

An overview of the framework is shown in Figure 2. VLWM is trained to predict goal description, goal interpretation, actions ($A$) and world state changes ($\Delta S$) – conditioned on visual context from past observations. It enables straightforward plan generation via text completion, using the proposed action directly as policy. We term this approach **system-1** planning.

However, the autoregressive nature of token decoding limits foresight and reflection, as each action decision become irreversible once made. Additionally, the trained policy might also clone suboptimal behaviors in the large-scale, real-world video data. We therefore introduce a reflective **system-2** *"planning with reasoning"* mode. In this mode, VLWM first generates multiple roll-outs based on action candidates (either proposed by itself or externally provided) and predicts resulting world states. We then search for the candidate action sequence that minimize a scalar *cost*, which is evaluated by a **critic** module that assess the desirability of candidate plans.

To train VLWM system-1, we first compress raw video into a hierarchical TREE OF CAPTIONS, then refines it into structured goal-plan descriptions using an LLM-based SELF-REFINE (Madaan et al., 2023). The model learns both a predictive world model ($S_t, A_t \rightarrow S_{t+1}$) and an action policy ($S_t \rightarrow A_{t+1}$) from this data.

The critic model in VWLM system-2 is a language model trained through a self-supervised objective: it learns to assign lower costs to valid progress toward the goal and higher costs to counterfactual or irrelevant actions in a constructed dataset, effectively measuring how closely each candidate action aligns with the desired goal state. The process of optimizing action plan by searching for a cost-minimizing candidate is a form of reasoning (LeCun, 2022). It enables the agent to perform trial-and-error internally with its learned world model to obtain the optimal action plans.

The VLWM is trained extensively on a large corpus of both web instruction videos and egocentric recordings, including COIN (Tang et al., 2019), CrossTask (Zhukov et al., 2019), YouCook2 (Zhou et al., 2018), HowTo100M (Miech et al., 2019), EgoExo4D (Grauman et al., 2024), EPIC-KITCHENS-100 (Damen et al., 2018). Collectively, there are 180k videos spanning over 800 days of duration. We generate TREE OF CAPTIONS for each video, resulting in a total of 21M nodes of unique detailed video captions (2.7 trillion words). With iterative LLM SELF-REFINE, we extracted

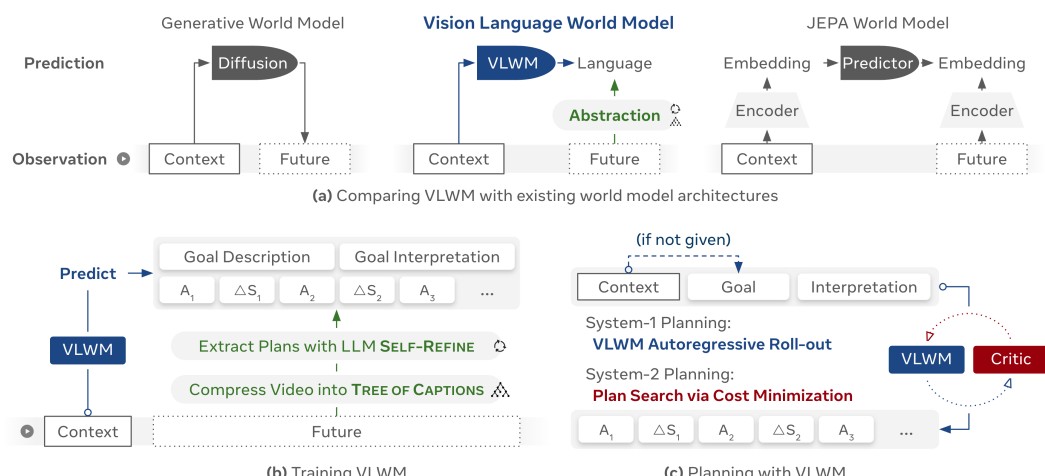

Figure 2: **Overview of VLWM. (a)** VLWM is a JEPA-style world model that predict abstract representation of future world states, instead of generating noisy and high-volume raw observations. **(b)** Given video contexts, VLWM's prediction target is a structured textual representation of the unobserved future. It includes goal and interleaved action ($A$) world state changes ($\Delta S$), all extracted automatically. **(c)** VLWM can infer possible goals from the context, and interpret them with current initial state and the expected final state. It supports both fast reactive system-1 plan generation and reflective system-2 reasoning based on cost minimization.

1.2 million trajectories of goal-plan pairs, consisting of 5.7 million steps of actions and states. We also reformulate text-only chain-of-thought reasoning paths in NaturalReasoning (Yuan et al., 2025) to action-state trajectories, obtaining additional 1.1 million goal-plan pairs.

Our evaluations cover both human ratings of plan preference, and quantitative results on various benchmarks. On the Visual Planning for Assistance (VPA) benchmark (Patel et al., 2023; Islam et al., 2024), VLWM achieved relative gains of +20% in SR, +10% in mAcc, and +4% in mIoU. Evaluated on human ratings in our proposed PLANNERARENA, the procedural plans generated by VLWM system-2 mode is more preferred than prompting based methods. On the RoboVQA benchmark (Sermanet et al., 2024), VLWM achieves 74.2 BLEU-1 score, outperforming strong VLM baselines. We further evaluate critic models for task completion, and our trained critic outperforms baseline semantic similarity models on both in-domain and OOD scenarios. It also established a state-of-the-art on WORLDPREDICTION (Chen et al., 2025) procedural planning task with 45% accuracy. Models and data will be open-sourced.

## 2 METHODOLOGY: VISION-LANGUAGE WORLD MODEL

We aim to train a world model that understands and predicts how actions affect physical world states, and to develop a framework for reasoning and planning in the real world where the world model serves as the core component. Our approach builds on the agent architecture introduced by LeCun (2022), where a reward-agnostic world model perform roll-out given candidate action plans, and the agent evaluates how closely each roll-out advances the current state toward the desired goal, and select the plan that minimizes this distance (*i.e.,* the cost).

In the sections below, §2.1 details how we extract structured language-based representation as future world state abstractions, which includes semantic compression techniques for efficiency considerations and quality optimization strategies. Then, §2.2 introduces how the critic is trained to evaluate cost in a self-supervised manner and explain the system-2 plan search based on cost-minimization.

### 2.1 VLWM SYSTEM-1

Given a video, VLWM system-1 aims to extract a structured language representation shown in Fig. 2 (b), which consists of a goal (description and interpretation) and a procedural plan (action-state sequence). For such a video-to-text extraction task, one straightforward approach would be to provide

a VLM with the full video and prompt it to extract the language representations. However, an important challenge arises: within a practical compute and memory budget, it is not feasible to simultaneously achieve 1) high spatial resolution for fine-grained perception, 2) long temporal horizon that spans many procedural steps, and 3) the use of a large and smart VLM that can follow complex instructions.

To address this challenge, we propose a two-stage strategy. First, the input video is compressed into a dense TREE OF CAPTIONS, which significantly reduces the data volume while preserving essential semantic information (§2.1.1). Then, structured goal-plan representations are extracted from these captions with LLMs. Since the second stage operates purely on text, it enables efficient processing with large LLMs and allows for iterative quality refinement through SELF-REFINE (§2.1.2).

### 2.1.1 COMPRESS VIDEO INTO TREE OF CAPTIONS

Each TREE OF CAPTIONS consists of a set of video captions generated independently from different local windows of a video, collectively forming a hierarchical tree structure. It aims to capture both fine-grained local details and long-horizon global information (Chen et al., 2024a). A key challenge lies in adaptively determining the tree structure, *i.e.,* the arrangement of different levels of windows for caption generation. Ideally, each node or leaf should correspond to a coherent *monosemantic* unit (Chen et al., 2024b), avoiding span across semantic boundaries. Existing temporal action localization and segmentation models (Ding et al., 2023) are limited in their openness, as they rely on human annotations with closed-vocabulary action taxonomies and are typically trained on narrow video domains.

We propose to create the tree structure via hierarchical feature clustering. Specifically, let $X$ be an untrimmed video, and let its feature stream be represented as $Z = \phi(X) = [\mathbf{z}_1; \ldots; \mathbf{z}_T] \in \mathbb{R}^{T \times d}$, where each $\mathbf{z}_t$ is a $d$-dimensional feature vector produced by a video encoder $\phi$. We segment the feature stream $Z$, and accordingly the underlying video $X$, using **hierarchical agglomerative clustering** (Murtagh & Contreras, 2012). Starting from the finest granularity—treating each item $\mathbf{z}_t$ as an individual cluster—the algorithm iteratively merges adjacent segments with the smallest increase in within-segment feature variance (*i.e.,* a measure of *polysemanticity*). This merging procedure is continued until there is only a single root node, and the full trace gives a hierarchical structure, where each node corresponds to a segment of the video.

The choice of $\phi$ determines the behavior of the segmentation. In this paper, we adopt the Perception Encoder (Bolya et al., 2025)–a state-of-the-art model that excels at extracting scene and action information from videos. Once the hierarchical tree structure is constructed, we generate detailed captions for each video segment, excluding short segments shorter than five seconds. We use PerceptionLM (Bolya et al., 2025) for detailed video captioning. The resulting TREE OF CAPTIONS achieves substantial compression: for instance, 1.1 TB video files in Ego4D (Grauman et al., 2022) can be compressed to under 900 MB of caption files.

### 2.1.2 EXTRACT PLANS WITH LLM SELF-REFINE

Given the compressed TREE OF CAPTIONS extracted from the video, our next objective is to derive a structured textual representation that serves as the prediction target for VLWM. This representation includes the following four components:

1. **Goal description** is a high-level summary of the overall task (*e.g., "cook tomato and eggs"*). In downstream applications, goal descriptions given by users are typically concise (*e.g.,* single sentence), omitting fine-grained details that define the final state. Therefore, explicit goal interpretations are required.

2. **Goal interpretation** includes contextual explanations that outlines both the initial and the expected final world states. The initial state describes the current status of tools, materials, and dependencies, etc., providing essential grounding for plan generation. The final state interprets the goal description concretely to facilitate cost evaluation in system-2 planning. For example, *"To achieve the goal, the eggs need to be cooked and mixed with tomatoes, and the mixture should be seasoned appropriately. The eggs should be whisked thoroughly to achieve a uniform texture..."*

3. **Action description** are the final outputs of the system that will be passed to downstream embodiments for execution or presented to users (*e.g., "Preheat the skillet on the stove"*). They must be clear, concise, and sufficiently informative to enable the receiver to understand and produce the intended world state transitions.

4. **World states** are internal to the system and serve as intermediate representations for reasoning and plan search. They should be a information bottleneck: sufficiently capturing all task-relevant consequences of actions while containing minimal redundancy. For example: *"This action prepares the skillet for cooking the eggs by increasing its temperature. The state of the skillet changes from cold to hot, making it ready for cooking. The oil used for preheating prevents the eggs from sticking to the skillet, ensuring they cook evenly..."*. See Appendix H.1 for more examples.

To ensure that the generated components meet these requirements, we adopt an iterative SELF-REFINE procedure (Madaan et al., 2023), leveraging LLMs as optimizers (Yang et al., 2023a). We begin by providing the LLM with detailed descriptions of the output requirements, examples of the expected format, and the formatted TREE OF CAPTIONS as input to generate an initial draft. In each refinement iteration, the LLM first provide a feedback to the current draft and produces a revised version accordingly. This self-refinement process is repeated for a predefined number of iterations, progressively optimizing output quality (see Appendix §G for examples).

To input TREE OF CAPTIONS to LLMs, we format it using a depth-first search (DFS) traversal order. This linearization aligns with the hierarchical structure of textual documents that LLMs are typically trained on and familiar with (*e.g.,* Section 1 → 1.1 → 1.1.1 → 1.1.2 → ...). In this paper, we use `Llama-4 Maverick` for its efficient inference and support for extended context length. Notably, the SELF-REFINE methodology is not tailored to specific LLM architecture.

### 2.1.3 VLWM SYSTEM-1 TRAINING

The training task of VLWM is defined in Eq.1. Here the `config` acts as system prompts. The `context` provides environmental information and can be either visual, textual, or both. The VLWM is trained to predict the future, represented by 1) `goal` description along with its interpretation (*i.e.,* the initial and expected final states), and 2) a `trajectory` consisting of sequence action ($A$) state ($\Delta S$) pairs. VLWM optimize the cross-entropy loss for next-token prediction on the right-hand side of Eq.1:

$$[\texttt{config}, \texttt{context}] \xrightarrow{\text{VLWM}} [\texttt{goal}, \texttt{interpretation}, \underbrace{\langle A_0, \Delta S_0 \rangle, \ldots, \langle A_N, \Delta S_N \rangle}_{\texttt{trajectory}}]. \quad (1)$$

This input-output formulation reflects three levels of world modeling: 1) contextual goal inference, the prediction of the possible future achievements, 2) action anticipation–proposing possible next actions, and 3) action-conditioned world state dynamics prediction. Since actions and resulting state changes are generated in an interleaved, autoregressive manner, it enables straightforward **System-1 Reactive Planning** through direct text completion. Given the `config`, `context`, and the `goal` description, VLWM interprets the goal and generates a sequence of action-state pairs until an `<eos>` token is reached. From a language modeling perspective, the world state descriptions act as internal chains of thought: they articulate the consequences of each action, allowing VLWM to track task progress and suggest appropriate next steps toward the goal. This planning mode is computationally efficient and well-suited for short-horizon, simple, and in-domain tasks.

Due to the (`video`, `text`) → `text` formulation in Eq.1, pretrained VLM can be used to initialize VLWM. This provides VLWM with strong visual perception, while also enabling it to inherit language understanding and generation capabilities, and commonsense knowledge in LLMs.

## 2.2 VLWM SYSTEM-2

While the System-1 mode allows fast plan generation, it lacks the capabilities of having foresight, evaluating of alternatives, or revising suboptimal decisions. Once an action is emitted, it is fixed, preventing the model from reconsidering or correcting errors. This reactive behavior can lead to

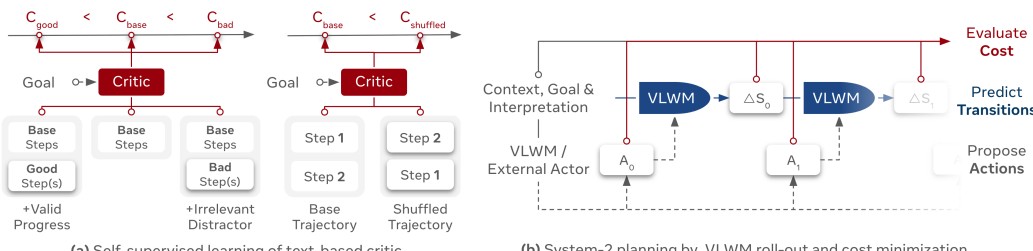

**Figure 3: System-2 planning of VLWM. (a):** the critic is trained in a self-supervised manner, assigning lower cost to valid progress, while assigning higher cost for adding irrelevant distractors or shuffling the steps. **(b):** VLWM generates candidate action sequences and simulates their future state transitions. A critic evaluates the resulting state trajectories given the goal, and the planner selects the lowest-cost plan.

error accumulation, particularly in long-horizon or complex tasks. To address these limitations, we introduce **System-2 Reflective Planning**, where the world model is coupled with a *critic module* that evaluates the desirability of multiple predicted futures given the goal.

### 2.2.1 PLANNING BY COST MINIMIZATION

System-2 planning involves the coordination of three components: the VLWM, the critic, and an actor. As illustrated in Fig. 3(b), the actor proposes candidate action sequences, the VLWM simulates their effects, and the critic evaluates their costs. The final plan is selected by identifying the candidate sequence with the lowest predicted cost.

The actor can be instantiated either as the VLWM itself or as an external module (*e.g.,* LLMs), particularly in cases where additional constraints on the action space or output format must be respected. The actor may vary the number of proposed candidates to control the search width or generate partial plans to enable more efficient tree search. In additional to the cost evaluated by the critic, task-specific penalties or guard-rails can be incorporated into the cost function, allowing the planner to respect external constraints, safety rules, or domain-specific preferences.

### 2.2.2 VLWM SYSTEM-2 TRAINING

In world model-based planning, the cost function typically quantifies the distance between the world state resulting from a candidate plan and the desired goal state (Zhou et al., 2024; Assran et al., 2025). It gives an estimation of how well the current task progress aligns with the intended goal and expected final state. In JEPA world models, this can be directly measured by *L1* or *L2* distance between fixed-dimensional embedding representations of world states. However, with VLWM, we must measure the **semantic distance** between language-based world state representations instead calculating distance in token space.

Formally, given VLWM predictions as described in Eq. 1, we aim to establish a distance function **critic** that evaluate cost $C = \textbf{critic}(\{\texttt{goal, interpretation}\}, \{\texttt{trajectory}\})$. Ideally, the cost should be *low* when the predicted trajectory reflects meaningful progress toward the goal, and *high* when it deviates due to irrelevant or erroneous actions. To model this behavior, we train a language model in a self-supervised manner, enabling it to assess the semantic quality of predicted plans without requiring explicit annotations. As shown in Fig. 3(a), we explore two types of self-supervised training signals for the critic:

1. We construct training samples by starting from a base partial trajectory and appending either (i) valid next step(s) resulting from a coherent continuation of the task, or (ii) distractor step(s) sampled from an unrelated task. The critic independently predicts three cost scores: $C_{\text{base}}$, $C_{\text{good}}$, and $C_{\text{bad}}$ and the model is trained to satisfy the ranking constraints $C_{\text{good}} < C_{\text{base}} < C_{\text{bad}}$, encouraging the critic to distinguish meaningful progress from irrelevant or misleading continuations.

2. We generate negative samples by randomly shuffling the steps in a base trajectory, producing a corrupted sequence with cost $C_{\text{shuffled}}$. The critic is then trained to enforce $C_{\text{base}} < C_{\text{shuffled}}$, ensuring sensitivity to procedural order and temporal coherence.

The critic is trained to minimize the following ranking loss with a fixed `margin`, supplemented with a cost centering regularization term weighted by a small constant $\lambda$ (Naik et al., 2024). To construct training pairs $\langle C_{\text{positive}}, C_{\text{negative}} \rangle$, we iterate over all three types of self-supervised signal described above: $\langle C_{\text{good}}, C_{\text{base}} \rangle$, $\langle C_{\text{base}}, C_{\text{bad}} \rangle$, and $\langle C_{\text{base}}, C_{\text{shuffled}} \rangle$.

$$\mathcal{L}_{\text{critic}} = -\max\left(0, \texttt{margin} + C_{\text{positive}} - C_{\text{negative}}\right)^2 + \lambda\left(C_{\text{positive}}^2 + C_{\text{negative}}^2\right). \tag{2}$$

## 3 Experiments

### 3.1 Implementation Details

**VLWM-8B.** We build training targets in two stages. First, raw videos from COIN, CrossTask, YouCook2, a filtered subset of HowTo100M (web instructions), and EPIC-KITCHENS-100 and EgoExo4D (egocentric) are compressed into a hierarchical Tree of Captions using `PE-G14` for features and `PerceptionLM-3B` for detailed window captions; ASR transcripts (web videos) and EgoExo4D expert commentary are included. We then extract structured {goal, interpretation, (action, $\Delta$state)} trajectories with two rounds of LLM Self-Refine (Llama-4 Maverick) on the linearized trees. VLWM is initialized from `PerceptionLM-8B` and trained as a conditional next-token predictor over the trajectory format (Eq. 1) with mixed video–text context: uniformly sampled 32 frames at $448^2$, max context 11.5k tokens, global batch 128. Training uses 12 nodes $\times$ 8 H100, and converges in $\sim$5 days. See App. Table 6 for a summarization of VLWM training data.

**VLWM-Critic-1B.** The critic takes (`goal+interpretation`, partial `trajectory`) and outputs a scalar cost used for system-2 search (Eq. 2). We construct self-supervised training pairs from our trajectories by (i) appending coherent next steps vs. distractors, and (ii) shuffling step order to create temporally inconsistent negatives; we additionally add preference triplets (UltraFeedback, Orca-DPO, Math-Step-DPO) and sentence-similarity triplets (MS-MARCO, SQuAD, HotPotQA, NQ, FEVER) by mapping {query, chosen/rejected} to {goal, positive/negative steps}. The critic is initialized from `Llama-3.2-1B`, trained for one epoch with batch 128, 1,536-token context, margin=1 and $\lambda$=0.01 in Eq. 2, on 1 node $\times$ 8 H100. Further details and ablations are in App. §B.2.

### 3.2 Visual Planning for Assistance (VPA)

**Evaluation Setup.** To verify that VLWM's large-scale pre-training yields practical gains in procedural planning, we adopt the Visual Planning for Assistance (VPA) benchmark (Patel et al., 2023). VPA measures how well a model can predict the next $T$ high-level steps of an ongoing activity given the video history and an explicit textual goal. We follow the standard evaluation horizons $T = 3$ and $T = 4$. Experiments are conducted on two widely used instructional-video corpora for procedual planning. COIN (Tang et al., 2019) contains 11 827 videos spanning 180 tasks, whereas CrossTask (Zhukov et al., 2019) comprises 2750 videos across 18 tasks. We adhere to the official train/val/test splits so results are directly comparable to prior work. VLWM is fine-tuned on the VPA training splits of COIN and CrossTask using the same hyper-parameters as in pre-training to match the target action vocabulary. Note that we use the "Protocol 2" (p2) (Wang et al., 2023) for evaluation, where the model cannot see any future actions. Instead, baseline models other than `PDPP` use p1 where the first 2 seconds of the next action is visible to the model.

**Results.** Table 1 confirms that VLWM sets a new state-of-the-art on the VPA benchmark. Across both COIN and CrossTask, and at both horizons $T = 3$ and $T = 4$, our model outperform existing baselines on most metrics. Compared to VidAssit which adopts a 70B LLM, our VLWM is much smaller (8B) while achieving superior results on 8/12 metrics. Averaged over the four settings, VLWM delivers absolute gains of +3.2% in SR, +3.9% in mAcc, and +2.9 points in mIoU.

### 3.3 PlannerArena

**Evaluation Setup.** We introduce PlannerArena, a human preference framework inspired by ChatbotArena (Chiang et al., 2024), where annotators compare anonymous model plans head-to-head (Figure 4); pairwise outcomes are converted to Elo ratings and win rates, aligning evaluation with practical assistant use. We test on COIN, CrossTask, and EgoExo4D, comparing VLWM

Table 1: **Visual Planning for Assistance** performances comparison against our finetuned VLWM.

| Model | COIN T=3 | | | COIN T=4 | | | CrossTask T=3 | | | CrossTask T=4 | | |
|---|---|---|---|---|---|---|---|---|---|---|---|---|
| | SR | mAcc | mIoU | SR | mAcc | mIoU | SR | mAcc | mIoU | SR | mAcc | mIoU |
| DDN (Chang et al., 2020) | 10.1 | 22.3 | 32.2 | 7.0 | 21.0 | 37.3 | 6.8 | 25.8 | 35.2 | 3.6 | 24.1 | 37.0 |
| LTA (Grauman et al., 2022) | - | - | - | - | - | - | 2.4 | 24.0 | 35.2 | 1.2 | 21.7 | 36.8 |
| VLaMP (Patel et al., 2023) | 18.3 | 39.2 | 56.6 | 9.0 | 35.2 | 54.2 | 10.3 | 35.3 | 44.0 | 4.4 | 31.7 | 43.4 |
| PDPP (Wang et al., 2023) | 21.6 | _47.2_ | _65.1_ | _16.2_ | **46.9** | _69.5_ | 11.6 | **36.7** | 47.7 | 6.3 | **35.1** | 50.9 |
| VidAssist (Islam et al., 2024) | _21.8_ | 44.4 | 64.4 | 13.8 | 38.3 | 66.3 | _12.0_ | 36.7 | **48.9** | **7.4** | 31.9 | **51.6** |
| VLWM-8B (ours) | **27.9** | **50.1** | **69.3** | **19.4** | _46.7_ | **74.0** | **13.5** | _36.4_ | _48.3_ | _7.2_ | _33.6_ | _51.1_ |

Figure 4: Illustration of PLANNERARENA.

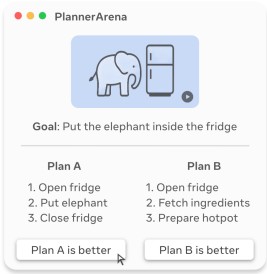

Goal: Put the elephant inside the fridge

Plan A
1. Open fridge
2. Put elephant
3. Close fridge

Plan B
1. Open fridge
2. Fetch ingredients
3. Prepare hotpot

[Plan A is better] [Plan B is better]

Table 2: **PLANNERARENA results.** Overall Elo score of our VLWM with a cost minimizing critic (VLWM System-2) and VLWM with a cost maximizing critic (*System-1 worst case), compared to other multimodal LLMs and ground truth plans.

| Model | # Parameters | Overall Elo Score | COIN | Win Rate (%) CrossTask | EgoExo4D |
|---|---|---|---|---|---|
| VLWM System-2 | 8B VLWM + 1B critic | **1261** | **87.9** | _70.6_ | **87.9** |
| Llama-4-Maverick | 400B | _1099_ | _66.7_ | **89.6** | 57.1 |
| VLWM System-1* | 8B VLWM | 992 | 34.3 | 37.0 | 50.0 |
| Qwen2.5VL | 72B | 974 | 38.2 | 34.8 | 18.3 |
| Ground Truth Plan | - | 952 | 43.6 | 42.2 | _69.5_ |
| PerceptionLM | 8B | 721 | 33.3 | 27.0 | 14.8 |

System-2 (search over 20 plans scored by an 8B cost-minimizing critic) and a cost-maximizing variant (System-1 worst-case) against leading VLMs and ground-truth plans. Battles are uniformly sampled across model pairs; all models start at Elo 1000 with K=32. Five annotators judged 550 pairs; three completed a 90-item pilot for agreement. See Appendix D for full details.

**Results.** Table 2 reports Elo and per-dataset win rates. VLWM System-2 leads with 1261 Elo; Llama-4-Maverick ranks second at 1099. The cost-maximizing VLWM (992 Elo) generally beats ground truth and models like Qwen2.5 and PerceptionLM. Ground-truth quality varies: EgoExo4D is relatively strong (GT win rate 69.5% vs. VLWM System-2 at 87.9%), while COIN and CrossTask GT trails (43.6% and 42.2%), underscoring limitations of current procedural planning datasets.

## 3.4 ROBOVQA

**Evaluation Setup.** To further assess VLWM's capabilities in grounded high-level reasoning and planning, we evaluate it on the RoboVQA benchmark (Sermanet et al. (2024)). RoboVQA challenges models to perform robotics-focused visual question answering in real world settings. We follow the standard evaluation protocols of RoboVQA and compare finetuned VLWM's performance using BLEU scores.

Table 3: Results on RoboVQA.

| Model | BLEU-1 |
|---|---|
| GPT-4V | 32.2 |
| RoboMamba-3B (Liu et al.) | 54.9 |
| PhysVLM-3B (Zhou et al., 2025) | 65.3 |
| ThinkAct (Huang et al., 2025) | 69.1 |
| RoboBrain-7B (Ji et al., 2025) | _72.1_ |
| VLWM-8B (ours) | **74.2** |

**Results.** Table 3 demonstrates that VLWM achieves highly competitive performance on the RoboVQA benchmark. Despite not being specialized on robotic data like some of the top-performing models such as RoboBrain, VLWM attains strong BLEU scores across all n-gram levels, ranking within the top two models. Notably, VLWM achieves the highest BLEU-4 score of 55.6, surpassing RoboBrain's 55.1, and closely follows it on BLEU-1 to BLEU-3. These results highlight VLWM's robust generalization and its ability to effectively integrate visual and language information for grounded reasoning and planning in embodied settings.

## 3.5 EVALUATION OF COST ESTIMATION

**Evaluation Setup.** We conduct intrinsic evaluations of the critic model independently of VLWM-8B roll-outs to assess whether it exhibits the intended behavior. Given a `goal` and a `trajectory` composed of a concatenation of $N_{gold}$ steps of reference plan that achieves the goal, and $N_{distractor}$ irrelevant steps appended after, the task asks the critic model to independently evaluate costs for every partial progress from the beginning, *i.e.,* $C_1 =$

Table 4: Goal achievement detection benchmark results.

| Model | VLWM | | | OGP | | |
|---|---|---|---|---|---|---|
| | Instruct | Ego | Overall | Robot | WikiHow | Overall |
| Chance Performance | 8.9 | 8.6 | 8.8 | 12.3 | 12.2 | 12.1 |
| Qwen3-Embedding-0.6B | 65.0 | 55.2 | 62.5 | 30.9 | 22.6 | 22.7 |
| Qwen3-Embedding-4B | 61.1 | 54.6 | 59.4 | 32.1 | 25.0 | 24.7 |
| Qwen3-Embedding-8B | 67.8 | 62.3 | 66.2 | 29.4 | 36.4 | 35.4 |
| Qwen3-Reranker-0.6B | 59.0 | 55.2 | 57.9 | 43.4 | 34.6 | 35.1 |
| Qwen3-Reranker-4B | 55.8 | 46.1 | 53.8 | 55.7 | 33.7 | 34.5 |
| Qwen3-Reranker-8B | 68.0 | 65.4 | 67.3 | 65.6 | 48.3 | 49.3 |
| **VLWM-critic-1B** | **98.4** | **92.7** | **96.9** | **72.9** | **48.3** | **50.0** |

Table 5: Ablation of goal interpretation or world state change descriptions in critic's input.

| Dataset | | Default | w/o Interp. | w/o states |
|---|---|---|---|---|
| Instruct | COIN | 97.1 | 96.4 (-0.7) | 91.4 (-5.7) |
| | CrossTask | 98.8 | 98.5 (-0.3) | 92.9 (-5.9) |
| | YouCook2 | 99.2 | 99.1 (-0.1) | 94.5 (-4.7) |
| Ego | EgoExo4D | 95.2 | 94.0 (-1.2) | 82.2 (-13.0) |
| | EK-100 | 90.1 | 88.6 (-1.5) | 64.0 (-26.1) |
| **Overall** | | **96.9** | **96.3 (-0.6)** | **88.1 (-8.8)** |

$\texttt{critic(goal, trajectory}[0:1])$, $C_1 = \texttt{critic(goal, trajectory}[0:2])$, $\ldots$, until $C_{N_{\text{gold}}+N_{\text{distractor}}} = \texttt{critic(goal, trajectory}[0:N_{\text{base}}+N_{\text{distractor}}])$. Since the distance to the goal should be the lowest after $N_{\text{gold}}$ steps of reference plan, we calculate the goal achievement detection accuracy according to whether $N_{\text{gold}} = \arg\min[C_1, ..., C_{N_{\text{gold}}+N_{\text{distractor}}}]$.

**Results.** We compare VLWM-critic-1B with Qwen3-Embedding models and Qwen3-Reranker models (Zhang et al., 2025) as baselines, which are state-of-the-art models for measuring semantic similarity. the cost is computed as $C = -\text{sim}\langle\texttt{goal, trajectory}\rangle$. Results are shown in Table 4. Our VLWM-critic-1B outperform baselines on most subsets by a large margin. We provide extended discussions and visualizations in Appendix C.

Table 5 provides an ablation of critic input representation using VLWM-critic-1B and the VLWM data. We tried to remove the goal interpretations which contains descriptions of current and expected final goal state, and state descriptions from the trajectory representation and leave actions only. We see both ablation leads to performance reduction on goal achievement detection, and the reduction on unseen OOD data (the Ego subset) is more severe, showing the importance of interpretation and world state description for effective generalization.

### 3.6 WORLDPREDICTION-PP

**Evaluation Setup.** The WORLDPREDICTION benchmark (Chen et al., 2025) is designed to evaluate high-level world modeling and procedural planning capabilities. In its procedural planning subset, each sample provides initial and final visual states alongside four candidate action plans, represented by video sequences. The task is to identify the correctly ordered sequence among counterfactual distractors. We evaluate our critic modules on WORLDPREDICTION-PP, following the evaluation protocol for Socratic LLMs in (Chen et al., 2025). These textual inputs were provided directly to our VLWM-critic models to compute costs for each candidate plan, selecting the option with the lowest cost.

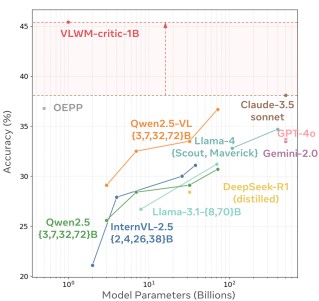

Figure 5: WorldPrediction-PP performance comparison.

**Results.** In Figure 5 (b), we compare our VLWM-critic models against baseline Socratic LLMs. Our VLWM-critic-1B established a new SoTA of 45.4% accuracy. Importantly, this evaluation constitutes a zero-shot scenario for VLWM-critic models, as neither the change captioning-based goal descriptions nor the detailed video captions as action steps were part of the training corpus.

## 4 CONCLUSION

We introduced VLWM, which predicts world dynamics in language space for interpretable and efficient high-level planning. Its dual-mode design supports fast System-1 planning and reflective System-2 planning via a self-supervised critic. Trained on large-scale instructional and egocentric videos, VLWM achieves state-of-the-art results on VPA, outperforms baselines in human preference studies, and ranks among top models on RoboVQA. These results highlight language-based world models as a powerful interface for bridging perception, reasoning, and planning, advancing AI assistants toward robust long-horizon decision making.

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

APPENDIX

This appendix supplements the main paper with: Related Works (§ A); Implementation Details for VLWM-8B and the 1B-parameter critic (§ B); Cost-estimation evaluation protocol and analyses (§ C); PLANNERARENA instructions, sampling, and interface notes (§ D); prompt templates for SELF-REFINE (§ E); and illustrative examples, including Tree-of-Captions (§ F), self-refinement feedback (§ G), and end-to-end VLWM planning rollouts (§ H).

## A    RELATED WORKS

### A.1    ACTION PLANNING

Planning is the task to generate a sequence of actions that can transit the world from initial state to a desired goal state. Our VLWM focuses on planning *high-level* actions, which is characterized by semantic and temporal abstraction (Sutton et al., 1999; Chen et al., 2025), as opposed to the low-level, high-frequency continuous actions in autonomous driving (Teng et al., 2023), robotics (Yu et al., 2020), and games (Mnih et al., 2015; Brockman et al., 2016), etc. Below, we compare existing methodologies for action planning.

**Imitation learning** (also known as behavior cloning) is effective when extensive expert demonstrations are available (Torabi et al., 2019; Baker et al., 2022). However, it becomes considerably more challenging when demonstrations are scarce or imperfect (Wu et al., 2019; Sagheb & Losey, 2025). For procedural planning and VPA tasks based on instructional videos (Chang et al., 2020; Patel et al., 2023), most approaches rely fundamentally on behavior cloning. Since the action annotations (Tang et al., 2019; Zhukov et al., 2019) are confined to limited vocabularies, the ground truth plans are frequently incomplete, making them not only suboptimal reference for benchmarking (which motivates our PlannerArena in §3.3), but also inadequate for imitation learning.

**Reinforcement learning** typically requires environments where agents can perform trial-and-error and receive explicit rewards. When environments support such interactions, reinforcement learning verifiable rewards (RLVR) is highly effective (Guo et al., 2025). Although RL is well-suited for domains where constructing simulation environments is viable, scaling RL to more diverse and complex domains is less feasible.

**Planning with reward-agnostic world model**. This approach exhibits superior generalization by learning from extensive, reward-free offline data (Sobal et al., 2025; Zhou et al., 2024; Assran et al., 2025). World models enable planning by simulating action outcomes internally and optimizing plans based on cost minimization. Unlike methods that predict task-specific rewards (*i.e.,* model-based RL (Hafner et al., 2024)), here world models only predict future world states (Ha & Schmidhuber, 2018), and action plans are optimized by minimizing the distance between the predicted resulting state and the desired goal state (LeCun, 2022). It allows inference-time scaling by conducting internal trial-and-error within the learned world model. Our VLWM's system 2 "planning with reasoning" leverages this paradigm, and we proved that it outperforms reactive system-1 behavior cloning.

### A.2    WORLD MODELING

World models aim to simulate environmental dynamics, enabling agents to optimize the plan without direct online interaction with the real environment. They have demonstrated success primarily in low-level control domains, such as autonomous driving (Wang et al., 2024b; Gao et al., 2024; Li et al., 2024) and robotics (Zhou et al., 2024), where models predict fine-grained, continuous sensory data over short horizons. Below, we compare existing world modeling approaches.

**Generative world models** typically utilize powerful diffusion-based architectures to reconstruct future observations directly (*e.g.,* in pixel space). Examples include Sora (Brooks et al., 2024), Cosmos (Agarwal et al., 2025a), Genie (Bruce et al., 2024; Parker-Holder et al., 2024) and UniSim (Yang et al., 2024), and recent multimodal chain-of-thought reasoning *i.e.,* "thinking with images" models (Su et al., 2025). While intuitive, generative models inherently suffer from computational inefficiency and task-irrelevant details entangled in pixel-based representations, severely limiting their

scalability for long-horizon planning. While these models generate realistic visuals, they have shown limited success in planning tasks.

**JEPA world models** encode observations into compact abstract representations, with a predictor trained to forecast these latent states. JEPA models have proven beneficial in representation learning, demonstrated by I-JEPA (Assran et al., 2023), IWM (Garrido et al., 2024), and V-JEPA (Bardes et al., 2024), and have facilitated MPC-based planning, exemplified by DINO-WM (Zhou et al., 2024), V-JEPA2 (Assran et al., 2025), and NWM (Bar et al., 2025). However, joint training of encoders and predictors poses challenges, notably the need for anti-collapse techniques such as EMA. Moreover, existing JEPA-based world models predominantly focus on low-level motion planning, and extending them to high-level action planning remains an open research challenge.

**Language-based world models** exploit natural language as a high-level abstraction interface, offering interpretability and computational advantages over pixel-based reconstruction. Prior work has explored prompting LLMs as world models (Hao et al., 2023; Tang et al., 2024; Wang et al., 2024a; Gu et al., 2024) or training language-based world model in narrowed domains, such as web navigation (Chae et al., 2024), text games (Lin et al., 2023; Wu et al., 2025), and in embodied environment (Wang et al., 2025a). In contrast, our VLWM approach explicitly learns a world model directly from large-scale raw video data.

## B  FULL IMPLEMENTATION DETAILS

### B.1  VLWM-8B

**Data.**  As summarized in Table. 6, the training videos for vision-language world modeling are sourced from two primary domains: 1) Web instruction videos: COIN (Tang et al., 2019), CrossTask (Zhukov et al., 2019), YouCook2 (Zhou et al., 2018), and a subset of HowTo100M (Miech et al., 2019) videos. These videos cover a diverse range of tasks, and provide clean expert demonstrations. 2) Egocentric recordings: EPIC-KITCHENS-100 (Damen et al., 2022) and EgoExo4D (Grauman et al., 2024). These videos feature continuous, uncut recordings in realistic wearable agent scenarios. For all datasets, we collect videos from their training split. While Ego4D (Grauman et al., 2022) is available as large-scale egocentric recordings dataset, we excluded it from training data to avoid potential overlap with benchmarks due to inconsistent train/val splitting.

We use Perception Encoder `PE-G14` (Bolya et al., 2025) and `PerceptionLM-3B` (Cho et al., 2025) ($320{\times}320$ spatial resolution, 32 frames per input – can be fit in 32GB V100) to generate the TREE OF CAPTIONS. We sample up to 5 target window per video according to the tree structure (the first 5 nodes in BFS traversal order), and use `Llama-4 Maverick` (mixture of 128 experts, 17B activated and 400B total parameters, FP8 precision) to extract plans from the window with the sub-tree of captions and two rounds of SELF-REFINE. Additional speech transcripts for web videos and the expert commentary in EgoExo4D are provided along with video captions to improve LLM's video understanding during plan extraction. In addition to video-based extraction, we repurposed the NaturalReasoning (Yuan et al., 2025) dataset to world modeling by replacing TREE OF CAPTIONS with the chain-of-thoughts. Action-state trajectories are extracted by LLM SELF-REFINE with similar prompts.

**Training**.  We use `PerceptionLM-8B` (Cho et al., 2025) to initialize our VLWM. The model is trained with a batch size of 128 and a maximum of 11.5k token context length. We perform uniform sampling of 32 frames in $448^2$ resolution for visual context inputs. With 12 nodes of $8{\times}$H100 GPUs, the training takes approximately 5 days.

### B.2  VLWM-CRITIC-1B

**Data.**  We generate paired data according to §2.2.2 from vision-language world modeling data of HowTo100M and NaturalReasoning. We also include TREE OF CAPTIONS data by by sampling subtrees and use root as goal and leafs as trajectories. We also incorporate off-the-shelf preference modeling data to train the critic, where the user queries are treated as goals and model responses are treated as actions. We derive $\langle C_{\text{positive}}, C_{\text{negative}} \rangle$ using $\langle$ "query" + "chosen" and "query" + "rejected" $\rangle$ . We include UltraFeedback (Cui et al., 2023), Orca DPO pairs (Lian et al., 2023), Math-Step-DPO (Lai et al., 2024) as sources of preference data. Lastly, we incorporate training data

Table 6: **Statics of VLWM data**. Vision-language world modeling data are extracted by generating TREE OF CAPTIONS from videos and performing iterative LLM SELF-REFINE. We combine six video sources and one text-only dataset.

| Domain | Additional Info | Dataset | # Videos (k) | Duration (hours) | # Trajectories (k) | # Steps (k) |
|---|---|---|---|---|---|---|
| Text-only | N/A | NaturalReasoning | - | - | 1,086.4 | 5,166.2 |
| Web Instruction Videos | ASR Transcripts | HowTo100M | 167.8 | 18,512.3 | 1,093.2 | 5,438.1 |
| | | COIN | 7.6 | 302.9 | 36.2 | 181.7 |
| | | CrossTask | 2.1 | 163.1 | 10.4 | 55.1 |
| | | YouCook2 | 1.2 | 102.3 | 5.8 | 31.9 |
| Egocentric Recordings | Expert Commentary | EgoExo4D | 0.6 | 53.5 | 3.1 | 18.8 |
| | N/A | EPIC-KITHCNES-100 | 0.5 | 68.9 | 2.2 | 14.0 |
| | Overall | | 179.8 | 19,202.9 | 2,179.6 | 10,604.3 |

for learning semantic similarity, where we convert triplets of `<query, positive sentence, negative sentence>` sentences to query as goal, positive sentence as positive action and negative sentence as negative action. This type of data includes MS-MARCO (Bajaj et al., 2016), SQUAD (Rajpurkar et al., 2016), HotPotQA (Yang et al., 2018), NaturalQuestions (Kwiatkowski et al., 2019), and FEVER (Thorne et al., 2018).

**Training.** The critic model is initialized from `Llama-3.2-1B` and trained for one epoch with a batch size of 128 (2.7k steps), maximum context length of 1536 tokens using a single node of 8×H100 GPUs. For hyper-parameters in Eq. 2, we set $\lambda$=0.01 and `margin`=1.

## C    DETAILS OF EVALUATION OF COST ESTIMATION

We construct testing sample from two sources. **1) Vision-language World Modeling (VLWM)**: 4,410 action-state trajectories extracted with TREE OF CAPTIONS and SELF-REFINE. The `goal` field combines both goal description and goal interpretation. Since VLWM-critic-1B is trained on HowTo100M trajectories, we exclude it and only sample data from other sources of instruction videos (COIN, CrossTask, YouCook2), and egocentric recordings (EgoExo4D, EPIC-KITCHENS-100). **2) Open Grounded Planning (OGP)**: Guo et al. (2024) released a collection of planning dataset containing goal-plan pairs sourced from different domains. We only use their "robot" subsets sourced from VirtualHoom and SayCan and WikiHow subset, since plans in the tool usage subset often contain too few number of steps. Different from VLWM data, trajectories in OGP only contain actions, and are OOD for both VLWM-critic-1B and baseline models. There are only 9,983 trajectories in OGP data.

In Figure 6, we visualize the normalized cost curves predicted by different critic models. The visualization can be viewed as "energy landscape", and the desired shape is to have the minimum cost at the 100% goal achievement point. On VLWM data, VLWM-ciritc-1B gives a much cleaner landscape compared to baselines. However, when comes to OGP datasets, the distribution becomes nosier. Despite domain gap and dataset noise problem mentioned above, one potential reasoning of performance degradation is the OGP gives action-only trajectory without any explicit world state descriptions, which makes cost evaluation harder.

## D    PLANNERARENA DETAILS

### D.1    INSTRUCTIONS & DATA

To evaluate model-generated plans, we conducted a controlled human evaluation study using a custom-built streamlit application. Annotators were presented with (i) a short video context, (ii) a textual goal (e.g., Make a fish curry), and (iii) two alternative plans generated by different anonymous models. The task is to select the preferred plan to achieve the goal given the provided video context. The instruction shown to annotators is:

> **PlannerArena Instruction**
>
> *You will see a short video that sets the context, then see a goal sentence. Two alternative plans (Plan A and Plan B) generated by a model are shown. Your job is to select the plan you would prefer to follow in order to achieve the stated goal within the given video context.*

The evaluation setup is based on three datasets commonly used for procedural video planning and understanding: COIN, CrossTask, and EgoExo4D. For all datasets, the video context given to the annotators is the entire original video truncated right before the start of the first annotated step in order to prevent models from leveraging future visual information in their plan. This is similar to the Visual Planning for human Assistance (VPA) setup, but in order to evaluate human plan preference. For EgoExo4D, the exo point of view is given as video context to prevent any partial observation problems.

We generate candidate plans with the other VLMs with zero-shot prompting, all models are provided with the same video context and were prompted with the following template:

```
You are provided with a context segment of a procedural video about
{goal_formatted}. Generate the remaining actions (steps) to take from
that context segment in order to reach the goal. The plan should be
composed of high-level descriptions starting with a verb, and it should
be clear and concise, including all essential information. There is no
need to be overly descriptive. Generate only the action steps.
```

## D.2 PAIRS SAMPLING & IAA

Unlike ChatbotArena which relies on an Elo-based sampling method to balance the evaluation across a large number of models, we adopt a uniform uniform sampling strategy as we only have six models to compare. Specifically, we first sample an equal number of battle pairs from each dataset, then enforce balanced participation across models such that each model competed equally against others within each dataset. A "setup" is defined as a (dataset, model pair) combination, and each setup is represented equally in the sample pool, yielding 3500 unique battle setups for PlannerArena.

Five annotators participated in the study. Prior to annotation, they completed a short warm-up consisting of five solved examples to familiarize themselves with the task. Inter-annotator agreement is computed over a shared subset of 100 samples with three annotators: the Fleiss' $K$ was 0.63, indicating substantial agreement, with a raw agreement percentage of 72.22%.

## D.3 EXAMPLE

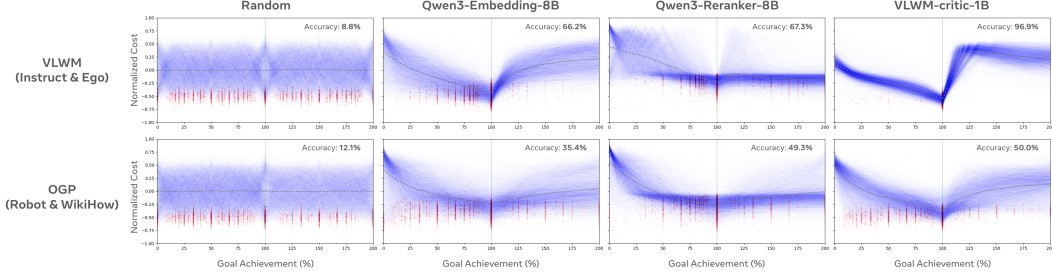

Figure 6: **Cost curves estimated by different critic models**. Each plot visualizes 3k cost curves on goal achievement detection trajectories, where each trajectory is composed of a reference gold plan (0%-100%) and distractor steps (100%-200%). Red dots (·) mark cost-minimizing steps (detected goal achievement points). VLWM-Critic accurately detects goal completion around 100% plan length, while baselines show suboptimal or noisy behavior.

*Instruction: You will see a short video that sets the context, then see a goal sentence. Two alternative plans (Plan A and Plan B) generated by a model are shown. Your job is to select the plan you would prefer to follow in order to achieve the stated goal within the given video context.*

**Goal: Build Simple Floating Shelves**

**Plan A**

1. cut shelve
2. assemble shelve
3. sand shelve
4. paint shelve
5. cut shelve
6. attach shelve

**Plan B**

1. cut the wooden planks to the desired length for the shelves
2. sand the cut edges to smooth them out
3. assemble the shelves by attaching the brackets to the wall
4. place the wooden planks onto the brackets
5. secure the shelves to the brackets using screws or nails
6. add decorative items to the shelves
7. test the stability of the shelves
8. make any necessary adjustments to the shelves

Which is the **better plan**?

● Plan A  ○ Plan B

Submit

Figure 7: PlannerArena interface. The sample shown here is from COIN, Plan A from the ground truth annotations and Plan B from Llama 4.

# E PROMPTS

## E.1 META PROMPT FOR LLM SELF-REFINE

---

```
{TREE OF CAPTIONS} {ADDITIONAL VIDEO INFO}

# Draft

Here is a draft for structured data extraction:

{PREVISOUS DRAFT}

---

# Your task

You carefully examine the draft above and identify problems. The
requirements are listed below. Go through each point one by one and
discuss aspects in the draft that doesn't meet the requirements. Be
specific and constructive, avoid vague and generic comments or simply
repeating the requirements. Quote the draft for detailed discussion.
Provide concrete points of explicit actionable revisions that could help
improve and enrich the draft. Make sure your revisions and the added
information is grounded in the provided content. After extensive
analysis and discussion, give the revision in the desired format.

{REQUIREMENTS OF PALN EXTRACTION}

# Output format

```yaml
discussion: |-
   Free form chain-of-thought reasoning: analyze the draft, identify
   problems, and suggest actionable revisions or enrichments.
plan:
- action: <action description>
  state: |-
     <world state change description and discussion>
  start: xx.xx # float between <min_start> and <max_end> round to two
  decimal digits
  end: xx.xx # float between <min_start> and <max_end> round to two
  decimal digits
- action: <action description>
  state: |-
     ...

goal: <goal description>
interpretation: |-
   <detailed goal interpretation>
```

Start your response with "```yaml\n..." and end with "\n```"
```

## E.2 REQUIREMENTS OF PLAN EXTRACTION FOR LLM SELF-REFINE

---

```
**Action Plan**

1. Identify a sequence of physical actions that meaningfully advance the
task progress; Omit vague, redundant, or purely presentational steps.
```

2. Each action is one informative imperative sentence said from the actor's perspective. Avoid describing actions from the tutor's or demonstrator's voice.
3. Infer the span of each action according the provided timestamps. They must fall within <min_start> and <max_end> and do not overlap with each other.
4. Be selective – time in the video may be non-linear. For example, the final result may appear at the beginning of the video. Such actions should be skipped.

**World State**
1. Explain how the action is performed according to the provided captions. Use imperative voice and instructional or tutoral style.
2. Provide elaborated discussion of the motivation, rationale, and purpose behind the action.
3. Discuss all relevant objects (can be both physical object or abstract concept, or the actor itself) whose states are changed by the action.
4. Cover various aspects, such as status, position, condition, temperature, etc. Highlight the causal relationship between actions and states.
5. Be logically coherent and semantically connected with neighboring steps. They are autoregressive and shouldn't conflict with each other.
6. Provide in-depth analysis. Perspectives may include (but are not limited to):
    * Implications of state changes
    * What the change enables; whether it is (or is not) ready for future steps
    * Whether and how the change advances or contributes to the overall goal
    * Whether it satisfy the desired final state for the activity, if not what is still required
7. Organize the discussion into a single coherent paragraph, it should be comprehensive and detailed, but also avoid redundancy and ensure readibility.

**Goal Identification**

1. Summarize the overall achievements by the actions during <min_start> to <max_end> (not the entire video).
2. Ensure comprehensive coverage. Feel free to use multiple sentences if appropriate.
3. Use imperative voice. But it should not be a simple concatenation of individual action names.
4. It summarize WHAT is achived (e.g., aggregation and abstraction of state changes) but not HOW it is achieved (e.g., "do x by doing y").

**Goal Interpretation**

1. Infer and describe the initial state of the environment before any action is taken. Only describe task-relevant aspects. Start with "Now, ..."
2. Interpret the goal in detail by discussing objects needs to be what state such that goal can be considered achieved.
3. Start with "To achieve the goal, ...". You can also include related technical specifications if applicable.
4. Description of the desired world state should be grounded in the provided context and aggregate all the state changes caused by the actions.
5. Discuss all objects, tools, materials, dependencies, etc. needed or invovled in the action steps and explain the functional rationale.
6. Use the tone as if you are now at the starting time of the video (<min_start>) and tasked to plan towards the given goal. You are preparing by thinking and analyzing the task.

7. Provide one paragraph and ensure its coherence and readibility. Importantly, you should avoid the leakage of any action plan information in this section.

**Overall Requirements**

1. Maintain faithfulness to the provided video content; Do not hallucinate or infer based on commonsense knowledge.
2. The output must strictly follow the given YAML format. Timestamps should be in the same format as <min_start> and <max_end>.
3. Except for the start and end times of the action, don't mention exact timestamp anywhere in your output.
4. Don't use 'the video' / 'the segment' in any part of the output. Instead, refer to the actions, objects, and environements directly.
5. Use specific functional description when referring to objects. Ignore task-irrelevant information such as appearance which does not affect the task.
6. Ensure comprehensiveness and detail in your output, but also conherence and readibility. Avoid repetition and redundancy.

# F    TREE-OF-CAPTIONS EXAMPLE

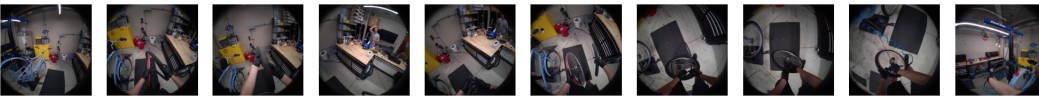

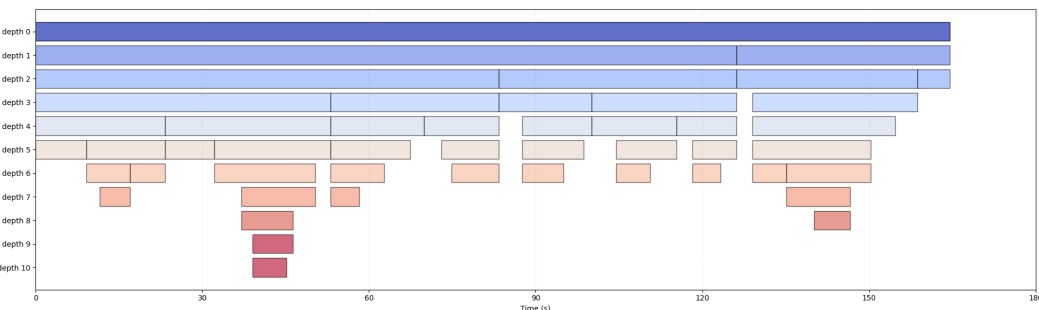

Figure 8: **Structure of TREE OF CAPTIONS (bottom) extracted from video (top)**. Each box is associated with a corresponding video caption.

**Tree-of-Captions formate by depth-first search (DFS)**:

```
# 0.00s -> 164.53s (duration: 164.5s)

The video features a view of a man repairing a bicycle tire and tube.
The man is wearing black gloves, and there is a bicycle lift holding a
blue bike in the background. In the background is another person wearing
a gray shirt. A black tool chest and a wooden tool bench can also be
seen ...

## Segment 1 - 0.00s -> 126.20s (duration: 126.2s)

This video features a man showing a second man how to repair a tire. The
second man stands center screen in a workshop. The man holding the tire
is center screen and behind him is a large workbench. In the background
are several other objects of various sizes and shapes ...

### Segment 1.1 - 0.00s -> 83.40s (duration: 83.4s)

The video features a first-person view of someone in a well-lit
workshop. The camera wearer is constantly seen touching objects in their
environment while working with a tire and a tube, which is later put on
a bicycle. The lighting is bright and focused. In the first shot, the
camera wearer holds a tire and a tube and feels the thread with their
hands, before passing the tire to the left hand ...

#### Segment 1.1.1 - 0.00s -> 53.13s (duration: 53.1s)

This video shows a busy workshop where two men are busy with their work.
The man in this video removes an inner tube from a bicycle wheel using a
tire lever, then peels off the tube using the thumb and fingers of his
right hand. He then puts the removed tube back on the table ...

**0.00s -> 23.27s (duration: 23.3s)**: The video features a person
working on a bicycle tire inside a bike shop. They wear black gloves and
pull a blue bicycle towards them. They take out a tire lever to remove
the tire from the wheel, and the focus is entirely on the cycle
maintenance process ...
```

```
**23.27s -> 53.13s (duration: 29.9s)**: The video is set inside a
workshop, where a bicycle is clearly visible at the left side of the
frame. The person's point of view is through a stere Pipeline video
camera mounted on a fishing rod, which caps off the end of his arm, so
the video looks like it is shot through a fish eye lens ...

#### Segment 1.1.2 - 53.13s -> 83.40s (duration: 30.3s)

The video shows a ...
```

## G  SELF-REFINE EXAMPLE

Below are some example feedback messages generated by `Llama-4 Maverick` during the SELF-REFINE process:

> *"Prepare the ingredients for Zucchini Curry."* in the draft could be **broken down into more specific actions** like *"Wash, peel, and chop the zucchini"* and *"Chop the onions and tomatoes."*

> The state change after sautéing the onions, ginger, garlic, and green chilies could include more details about **how this step affects the overall flavor and texture** of the curry.

> *The action of "Display the Zucchini Curry in a bowl"* is more of a **presentational step rather than a meaningful action** that advances the task progress, so it should be removed from the steps.

# H    VLWM Planning Examples

## H.1    Full Trajectory

```
<!------------ VLWM Input ------------>

<GOAL>
Cooking Tomato and Eggs
</GOAL>
```

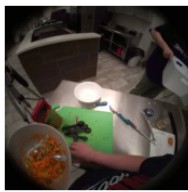 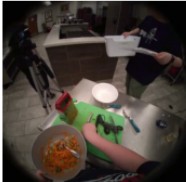 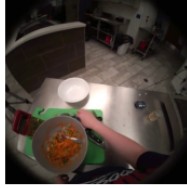 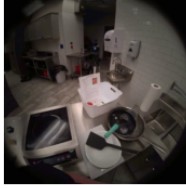 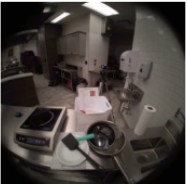

```
<!------------ VLWM Output ------------>

<INTERPRETATION>
Now, the kitchen is set up with necessary ingredients and tools. To
achieve the goal, the eggs need to be cooked and mixed with tomatoes,
and the mixture should be seasoned appropriately. The eggs should be
whisked thoroughly to achieve a uniform texture. The tomatoes should be
heated until they are soft and well coated with the eggs. The dish
should be seasoned with salt and pepper to taste. The final state of the
dish should be a well-cooked, flavorful mixture of eggs and tomatoes,
ready for serving. The kitchen environment and the tools used, such as
the stove, pan, and utensils, are crucial for achieving this goal. The
ingredients, including eggs, tomatoes, oil, salt, and pepper, are
essential for creating the desired dish.
</INTERPRETATION>

---

<ACTION>
Preheat the skillet on the stove
</ACTION>

<STATE>
To preheat the skillet, turn on the stove to medium-high heat and wait
for the skillet to heat up. The skillet is garnished with a little bit
of oil to prevent the eggs from sticking. This action prepares the
skillet for cooking the eggs by increasing its temperature. The state of
the skillet changes from cold to hot, making it ready for cooking. The
oil used for preheating prevents the eggs from sticking to the skillet,
ensuring they cook evenly.
</STATE>

---

<ACTION>
Crack eggs into a bowl and whisk them together
</ACTION>

<STATE>
Crack eggs into a bowl by breaking the eggshells over the edge of the
bowl and pouring the contents into the bowl. Use a spatula to mix the
eggs in a circular motion, being careful not to over-beate them. This
action changes the state of the eggs from whole to a mixed liquid,
making them ready for cooking. The eggs are now in a state that can be
seasoned and cooked. The whisking action ensures the eggs are well
mixed, contributing to a uniform texture in the final dish.
```

```
1404    </STATE>
1405
1406    ---
1407
1408    <ACTION>
        Season the eggs with salt
1409    </ACTION>
1410
1411    <STATE>
1412    Add salt to the eggs and mix well using a spatula. This action seasons
1413    the eggs, enhancing their flavor. The state of the eggs changes from
        unseasoned to seasoned, making them ready for cooking. The seasoning
1414    adds flavor to the eggs, which is crucial for the overall taste of the
1415    dish.
1416    </STATE>
1417
1418    ---
1419
1420    <ACTION>
        Cook the eggs in the pan
1421    </ACTION>
1422
1423    <STATE>
1424    Pour the egg mixture into the preheated pan with oil. Use a spatula to
        stir the eggs, breaking them into curds. Continue cooking until the eggs
1425    are almost set. This action changes the state of the eggs from liquid to
1426    a partially cooked state. The eggs are now partially cooked and are
1427    being transformed into curds. The heat from the pan cooks the eggs, and
1428    the stirring action ensures even cooking.
        </STATE>
1429
1430    ---
1431
1432    <ACTION>
1433    Remove the cooked eggs from the pan
        </ACTION>
1434
1435    <STATE>
1436    Use a spatula to scrape the cooked eggs out of the pan and onto a plate.
1437    This action removes the eggs from the pan, making way for cooking other
        ingredients. The state of the eggs changes from being in the pan to
1438    being on a plate. The eggs are now removed from the pan, allowing for
1439    the addition of other ingredients to be cooked.
1440    </STATE>
1441
1442    ---
1443
1444    <ACTION>
        Cook the tomatoes in the pan
1445    </ACTION>
1446
1447    <STATE>
1448    Add tomatoes to the pan and stir them using a spatula. Continue cooking
        until the tomatoes are soft. This action changes the state of the
1449    tomatoes from raw to cooked, making them ready for mixing with the eggs.
1450    The tomatoes are now cooked and can be combined with the eggs. The heat
1451    from the pan cooks the tomatoes, and the stirring action ensures even
1452    cooking.
1453    </STATE>
1454
1455    ---
1456    <ACTION>
1457    Add the cooked eggs back into the pan with the tomatoes
        </ACTION>
```

```
<STATE>
Add the cooked eggs back into the pan with the tomatoes. Use a spatula
to stir the mixture, combining the eggs and tomatoes. This action mixes
the eggs and tomatoes together, creating a uniform dish. The state of
the mixture changes from separate components to a combined dish. The
eggs and tomatoes are now mixed together, creating a cohesive and
ready-to-serve dish.
</STATE>

---

<ACTION>
Season the dish with salt and pepper
</ACTION>

<STATE>
Add salt and pepper to the dish and stir well using a spatula. This
action seasons the dish, enhancing its flavor. The state of the dish
changes from unseasoned to seasoned, making it ready for serving. The
seasoning adds flavor to the dish, ensuring it is palatable.
</STATE>

---

<ACTION>
Transfer the dish to a serving plate
</ACTION>

<STATE>
Use a spatula to transfer the cooked dish from the pan to a serving
plate. This action changes the state of the dish from being in the pan
to being on a plate, making it ready for serving. The dish is now
presented in a serving plate, ready to be consumed.
</STATE>

---

<ACTION>
Present the final dish
</ACTION>

<STATE>
The final dish is presented on a white surface. The dish is now ready to
be served. The state of the dish changes from being in the cooking
vessel to being on a serving plate.
</STATE>

---

<GOAL_ACHIEVED>
```

## H.2 COST MIN/MAXIMIZING PLANS

```
### Goal: Make Beef Tacos

Cost Minimizing Plan (cost = -4.86):
  1. Saute chopped onions and garlic in oil
  2. Add ground beef to the pan and break it up
  3. Add seasoning to the beef mixture
  4. Add tomato sauce to the beef mixture and stir
  5. Prepare taco shells
  6. Fill taco shells with the beef mixture
```

```
  7. Add cheese to the tacos
  8. Serve the tacos

Cost Maximizing Plan (cost = 3.03):
  1. Add959 existing ingredients to the slow cooker
  2. Mix the ingredients in the slow cooker
  3. Add taco seasoning to the slow cooker
  4. Stir the contents of the slow cooker
  5. Open the slow cooker lid
  6. Serve the cooked beef mixture onto a tortilla
  7. Add cheese to the taco
  8. Fold the tortilla

### Goal: Make Kerala Fish Curry

Cost Minimizing Plan (cost = -4.36):
  1. Dry roast the chilies in a pan over medium heat
  2. Add ginger and garlic to the pan and mix with the chilies
  3. Add water and tamarind paste to the pan and mix
  4. Add salt and sugar to the curry and stir
  5. Add fish to the curry and cook until done
  6. Add water to the curry and stir
  7. Simmer the curry and stir occasionally
  8. Garnish with cilantro
  9. Serve the curry

Cost Maximizing Plan (cost = -0.04):
  1. Dry red chili
  2. Add mustard seeds
  3. Saute the onions
  4. Add turmeric powder
  5. AddTransition metal pieces
  6. Add significance
  7. Add curry leaves
  8. Add chili powder
  9. Add coriander powder
  10. Add fish
  11. Add salt
  12. Add tamarind juice
  13. Finish with coconut oil

### Goal: Use Tapping Gun

Cost Minimizing Plan (cost = -7.58):
  1. Insert the gun head into the Tapping gun
  2. Remove the existing gun head from the Tapping gun
  3. Handle the Tapping gun to prepare it for a new gun head
  4. Clean the Tapping gun
  5. Prepare the Tapping gun for use by loading it with a new gun head
  6. Load the Tapping gun with a new gun head
  7. Demonstrate the final preparation of the Tapping gun
  8. Emphasize the completion of the task
  9. Conclude the demonstration
  10. Fill the Tapping gun with disposable scoops

Cost Maximizing Plan (cost = 3.46):
  1. Ignited the lighter to produce a flame
  2. Insert the gun head into the tag brand
  3. Load the Tapping gun with degrees
  4. Insert the Tapping gun into the red compartment
  5. Finish up by Aer
  6. Aer.YEAR4 doesn't come off easily

### Goal: Replace SIM Card
```

```
Cost Minimizing Plan (cost = -5.43):
  1. Eject the SIM card tray using a SIM card ejector tool
  2. Remove the SIM card from the tray
  3. Place the SIM card into the tray
  4. Reinsert the SIM card tray into the phone
  5. Ensure the SIM card is properly seated and the tray is fully inserted
  6. Adjust and finalize the SIM card tray insertion
  7. Verify the SIM card tray is fully inserted and functional

Cost Maximizing Plan (cost = 4.24):
  1. Use the SIM card ejector tool to eject the SIM card tray
  2. Remove the SIM card tray from the phone
  3. Place the SIM card into the SIM card tray
  4. Insert the SIM card tray back into the phone
  5. Place the phone on a flat surface
  6. Display the final result or a related message

### Goal: Use Rice Cooker To Cook Rice

Cost Minimizing Plan (cost = -4.31):
  1. Take out some rice
  2. Wash the rice
  3. Put the washed rice into the rice cooker
  4. Measure the water level
  5. Cook the rice using the rice cooker
  6. Open the rice cooker and fluff the cooked rice
  7. Serve the cooked rice

Cost Maximizing Plan (cost = 3.91):
  1. Take out some rice
  2. Rinse the rice
  3. Place the strainer over the rice cooker
  4. Add water to the rice cooker
  5. Start the rice cooker
  6. Analytical balance assessment
  7. Open the rice cooker to check the cooked rice
  8. Scoop the cooked rice from the rice cooker

### Goal: Make Chinese Spring Rolls

Cost Minimizing Plan (cost = -4.44):
  1. Pour oil into a wok and heat it
  2. Add chopped garlic and ginger to the heated oil in the wok
  3. Cook chopped vegetables such as carrots and onions in the wok
  4. Add noodles and various sauces to the wok and stir-fry
  5. Prepare spring roll sheets and fill them with the cooked mixture
  6. Fry the filled spring rolls in hot oil until they are golden brown
  7. Serve the fried spring rolls on a plate

Cost Maximizing Plan (cost = -0.59):
  1. Pour vegetable oil into a wok
  2. Add chopped garlic and ginger to the wok
  3. Stir-fry chopped onions
  4. Add chopped carrots and capsicum to the wok
  5. Add noodles and purified water to the wok
  6. Drain the cooked noodles
  7. Mix the cooked noodles with chopped parsley, salt, and pepper
  8. Prepare the cooking vessel for frying
```

## THE USE OF LARGE LANGUAGE MODELS

We used large language models (LLMs) solely as writing assistants for this paper. Specifically, they were employed to help rephrase sentences for clarity and readability. No content, ideas, or

experimental results were generated by LLMs. The authors take full responsibility for the scientific contributions and all written content.

