# OpenReview forum: "Planning with Reasoning using Vision Language World Model"
_ICLR.cc/2026/Conference — ICLR 2026 Conference Withdrawn Submission_

### Official Review · Reviewer_txpj · 2025-10-23

**Soundness:** 2
**Presentation:** 2
**Contribution:** 2
**Rating:** 6
**Confidence:** 3

**Summary:**

This paper introduces VLWM (Vision Language World Model), a foundation model that uses language as an abstract representation for high-level world modeling and planning. Trained on large-scale instructional and egocentric videos, VLWM predicts future actions and state changes in natural language, enabling both fast, reactive planning (System-1) and reflective, reasoning-based planning (System-2). The latter uses a self-supervised critic model to evaluate and select optimal plans by minimizing a semantic cost function. VLWM achieves state-of-the-art results on visual planning benchmarks, outperforms leading models in human preference evaluations, and demonstrates strong generalization in robotics and procedural reasoning tasks—all while being more efficient and interpretable than pixel-based world models.

**Strengths:**

The main strenghts of the paper are the following:
- An effective use of text as a compact, interpretable representation for world modeling, avoiding the inefficiency of pixel-based prediction.

- Combines fast, reactive planning (System-1) with a powerful, reflective reasoning mode (System-2) that searches for optimal plans via cost minimization.

- Achieves good performance on multiple benchmarks (VPA, RoboVQA, WorldPrediction) and wins in human preference evaluations (PlannerArena), demonstrating practical superiority on the considered scenarios.

- Leverages massive, diverse video datasets (180k videos) without needing task-specific rewards, enabling broad generalization.

**Weaknesses:**

The main weaknesses of the paper are:
- System-2 planning seem to require multiple model roll-outs and a search process, making it significantly more computationally expensive than System-1.

- The quality of training data relies heavily on an iterative LLM "Self-Refine" process, making the pipeline complex and potentially sensitive to the chosen LLM's capabilities.

- The critic model's performance degrades on out-of-distribution data, especially when world state descriptions are absent, highlighting a generalization gap that is not properly discussed and experimented.

- System-1 can clone suboptimal behaviors present in the large-scale, real-world training data.

- The paper lacks of proper detailed descriptions to allow and external individual to reproduce the results. The appendix adds some additional details, but they are not sufficient IMHO.

- The paper is not self contained, to analyze the positioning w.r.t. the state-of-the-art (related work) one has to read the appendix, not part of the paper itself.

**Questions:**

Q1: The paper notes System-2's performance gain but doesn't quantify its computational cost. Could you provide a direct comparison, such as the average wall-clock time or FLOPs required for System-2 to generate a single plan versus System-1? This is critical for assessing the practical trade-off between performance and efficiency.

Q2: The heavy reliance on a powerful, proprietary LLM (Llama-4 Maverick) for the "Self-Refine" data generation creates a significant bottleneck and reduces the accessibility and transparency of your method. How sensitive is the final VLWM performance to the quality and capability of the LLM used in this refinement stage? Have you experimented with smaller, open-source LLMs to test this dependency?

Q3: The observed performance drop on OGP data, especially without world states, is a major limitation that isn't sufficiently discussed. This suggests the critic relies on stylistic cues from its training data rather than robust reasoning. What specific steps are you exploring to improve the critic's generalization, such as different training data mixtures or architectural changes, to make it truly robust to novel goal-plan representations?

Q4: Since VLWM is trained on uncurated real-world videos, System-1 inherently clones demonstrated behaviors, including errors. Beyond using System-2 to correct them, does your framework have any inherent mechanism during training to identify or down-weight suboptimal trajectories in the dataset to improve the base policy?

Q5: While the appendix adds information, key details for reproduction are missing. For instance, the specific hyperparameters (learning rate, optimizer), the exact architectural modifications from PerceptionLM, and the code for the hierarchical clustering and plan search are not provided. Will you release the full training code and configuration files to ensure the community can verify and build upon these results?

Q6: Burying the "Related Work" section in the appendix severely undermines the paper's ability to stand on its own. It forces a reader to search for the authors' positioning within the existing literature, which is a fundamental part of evaluating a paper's novelty. Why was this critical section not included in the main body to provide immediate context for your contributions against world models like JEPA, Genie, and planning methods like PDPP? Notice that, the trivial answer of space limitations is unacceptable!

---

### Official Review · Reviewer_xJqX · 2025-10-30

**Soundness:** 4
**Presentation:** 3
**Contribution:** 3
**Rating:** 6
**Confidence:** 3

**Summary:**

This paper proposes VLWM, a vision-language world model that represents future states and actions in natural language. Using LLM-based annotation techniques such as Tree of Captions and Self-Refine, large-scale video data is converted into structured goal descriptions and interpretations. Two reasoning schemes are introduced: System-1 (autoregressive prediction) and System-2 (critic-guided rollouts). VLWM achieves new state-of-the-art results on multiple benchmarks, highlighting the promise of language-based state modeling for world models.

**Strengths:**

1. The paper introduces Tree of Captions and Self-Refine techniques to annotate large-scale video data with structured goal descriptions, goal interpretations, and action–state transitions. This pipeline and the curated dataset are valuable contributions that can significantly benefit the research community.
2. Building on VLWM, the paper further introduces a critic model as a selector for search results, which enhances the potential of world-model-based planning. Moreover, the critic is trained in a self-supervised manner, enabling scalability to large-scale training.

**Weaknesses:**

1. The paper introduces two self-supervised training schemes for the critic model — enforcing good < base < bad and base < shuffled. However, the paper does not clarify which pairing mode contributes more to training effectiveness, nor the ratio between different pair types during training. This lack of detail makes it difficult to assess how the critic’s performance is influenced by the design choices.
2. Clarity of exposition: Some key concepts are not explained with sufficient clarity. For example, in Section 2.1.1 the discussion of the Tree of Captions lacks explicit references to the illustrative examples in the appendix, which significantly hinders readers’ conceptual understanding during the reading process.
3. The relationship between System-1 and System-2 is not clearly explained in the evaluation. For example, in Table 2 only System-2 results are reported, but it is unclear what the corresponding System-1 results would be, or how large the performance gap is between the two across different tasks. Moreover, it is ambiguous whether the default VLWM-8B refers to the System-1 or System-2 variant. Clarifying these points would help readers better understand the distinctions and relative advantages of the two systems.

**Questions:**

See weakness.

---

### Official Review · Reviewer_GmGb · 2025-10-31

**Soundness:** 3
**Presentation:** 3
**Contribution:** 2
**Rating:** 4
**Confidence:** 4

**Summary:**

This paper presents the Vision Language World Model (VLWM), a foundation model for language-based world modeling on natural videos. VLWM infers high-level goals from visual inputs and predicts interleaved actions and state transitions, extracted through LLM self-refinement over compressed futures represented as a Tree of Captions. It jointly learns an action policy and a dynamics model, supporting fast system-1 plan decoding and reflective system-2 planning guided by a self-supervised semantic cost critic. VLWM sets new state-of-the-art results on Visual Planning for Assistance and PlannerArena (system-2 improving Elo by 27%), and surpasses strong baselines on RoboVQA and WorldPrediction benchmarks.

**Strengths:**

1. New Data Generation Pipeline: The two-stage pipeline is a clever and scalable method for creating structured, high-level plan data from uncurated videos.

2. Language as an Abstract World State: Using language as the world model's representation space is a powerful and timely direction, offering inherent interpretability and semantic abstraction for complex planning.

3. Formalization of System-1/System-2 Planning: The paper's explicit implementation of "system-1"  and "system-2" is an elegant framework, supported by empirical results.

**Weaknesses:**

1. The System-2 planner is ad-hoc, stitching together two separately trained models: a VLWM (planner) and a text-only LM (critic). There is no joint training, questioning if the critic is a generic text-ranker rather than a value function grounded in the VLWM's world model.

2. The core claim (system-2 > system-1) rests on a new, small-scale (550 pairs) human evaluation. This is insufficient for a major claim. The fact that the dataset's "groundtruth" (952 Elo) scores worse than the system-1 model (992 Elo) also questions the benchmark's validity.

3. The SOTA claim on VPA is misleading. VPA is a next-step prediction task that only evaluates the System-1 policy, not the system-2 planning and reasoning framework, which is the paper's main contribution. The lack of a System-2 evaluation on VPA is a major omission.

**Questions:**

All of my qeustions are listed in the weakness section. If my concerns are well addressed, I will raise my rating.

---

### Official Review · Reviewer_7fC9 · 2025-11-04

**Soundness:** 2
**Presentation:** 2
**Contribution:** 2
**Rating:** 2
**Confidence:** 2

**Summary:**

The paper introduces a Vision-Language World Model (VLWM) built as a two-system pipeline: (i) a compression + extraction + autoregressive (AR) “system-1” that converts videos into hierarchical text (Tree-of-Captions), extracts targets with SELF-REFINE, and trains a VLM to emit interleaved ⟨action, Δstate⟩ sequences; and (ii) a learned scalar critic (“system-2”) that scores candidate plans via a margin-ranking style objective, enabling selection of the lowest-cost rollout. Empirical results are reported on VPA, a human preference study (PlannerArena), RoboVQA, and critic-only tasks.

**Strengths:**

- Clean systems decomposition (compression → target extraction → AR training; critic-guided search) with intuitive figures and readable textual plans/states.

- Substantial engineering to assemble a large data/compute pipeline; end-to-end training at meaningful scale.

- Specialized critic is effective: ablations indicate that including state text improves critic performance; critic-only evaluations are well chosen for its role.

- Human-readable artifacts (goals, interpretations, ⟨action, Δstate⟩) improve interpretability and error analysis.

**Weaknesses:**

- Methodological novelty is limited: most components (hierarchical clustering, captioning, SELF-REFINE, AR cross-entropy training, margin-ranking critic) are standard; the contribution is primarily pipeline integration rather than a new planning/world-model objective.

- Framing exceeds the math/objectives: training is essentially language-space AR modeling plus a learned text critic; claims aligned with JEPA/latent prediction or a grounded world model are not substantiated by the presented objectives/constraints.

- Planning validation gap: no closed-loop or interactive evaluation; critic-only and offline metrics don’t establish that predicted Δstates guide effective action sequences under uncertainty or partial observability.

**Questions:**

Please confirm Equation (squared-hinge with a leading minus sign): is the optimization intended to minimize a negative squared violation term? If so, why doesn’t this reward larger violations? If it’s a typo, provide the exact loss used and update all results accordingly.

---

### Note · Authors · 2025-11-24

**Comment:**

Dear ICLR Program Committee and Reviewers,
We would like to formally withdraw our paper from consideration at ICLR. We sincerely appreciate the time and effort the reviewers dedicated to evaluating our work. Their insightful comments and suggestions have provided us with valuable perspectives that will help us improve our research.
We will carefully consider all the feedback as we continue to develop and refine our work. Thank you again for your thoughtful reviews and support.

**Withdrawal Confirmation:**

I have read and agree with the venue's withdrawal policy on behalf of myself and my co-authors.